# Correlation between Coronary Artery Calcium- and Different Cardiovascular Risk Score-Based Methods for the Estimation of Vascular Age in Caucasian Patients

**DOI:** 10.3390/jcm11041111

**Published:** 2022-02-19

**Authors:** Milán Vecsey-Nagy, Bálint Szilveszter, Márton Kolossváry, Melinda Boussoussou, Borbála Vattay, Béla Merkely, Pál Maurovich-Horvat, Tamás Radovits, János Nemcsik

**Affiliations:** 1MTA-SE Cardiovascular Imaging Research Group, Heart and Vascular Center, Semmelweis University, 1122 Budapest, Hungary; szilveszter.balint@gmail.com (B.S.); marton.kolossvary@gmail.com (M.K.); melinda.b.md@gmail.com (M.B.); bori.vattay@gmail.com (B.V.); maurovich.horvat@gmail.com (P.M.-H.); 2Cardiology Department, Heart and Vascular Center, Semmelweis University, 1122 Budapest, Hungary; szivct@gmail.com (B.M.); radovitstamasd@yahoo.com (T.R.); 3Medical Imaging Centre, Semmelweis University, 1082 Budapest, Hungary; 4Department of Family Medicine, Semmelweis University, 1085 Budapest, Hungary; janos.nemcsik@gmail.com; 5Health Service of Zugló (ZESZ), 1148 Budapest, Hungary

**Keywords:** vascular age, coronary artery calcium score, Framingham, SCORE

## Abstract

Vascular age can be derived from cardiovascular (CV) risk scores such as the Framingham Risk Score (FRS) and the Systematic Coronary Risk Evaluation (SCORE). Recently, coronary artery calcium score (CACS) was proposed as a means of assessing arterial age. We aimed to compare these approaches for the assessment of vascular age. FRS-, SCORE-, and CACS-based vascular ages of 241 consecutive Caucasian patients undergoing coronary CT angiography were defined according to previously published methods. Vascular ages based on FRS, SCORE, and CACS were 68.0 (IQR: 55.0–82.0), 63.0 (IQR: 53.0–75.0), and 47.1 (IQR: 39.1–72.3) years, respectively, (*p* < 0.001). FRS- and SCORE-based biological age showed strong correlation [ICC: 0.91 (95%CI: 0.88–0.93)], while CACS-based vascular age moderately correlated with FRS- and SCORE-based vascular age [ICC: 0.66 (95%CI: 0.56–0.73) and ICC: 0.65 (95%CI: 0.56–0.73), respectively, both *p* < 0.001)]. Based on FRS, SCORE, and CACS, 83.4%, 93.8%, and 42.3% of the subjects had higher vascular age than their documented chronological age (FRS+, SCORE+, CACS+), and 53.2% of the FRS+ (107/201) and 57.1% of the SCORE+ (129/226) groups were classified as CACS-. Traditional risk equations demonstrate a tendency of overestimating vascular age in low- to intermediate-risk patients compared to CACS. Prospective studies are warranted to further evaluate the contribution of different vascular age calculations to CV preventive strategies.

## 1. Introduction

Cardiovascular diseases (CVDs) are still the leading cause of global mortality and a major contributor to disability [1]. Precise assessment of CV risk remains the cornerstone of decision-making in the context of primary prevention and treatment management [2]. Conventional methods expressing CVD risk, however, yield substantial shortcomings that limit their utility in clinical practice [3,4]. Absolute cardiovascular risk is a statistical and epidemiological approach that most patients may consider hard to comprehend and may, therefore, impair patient motivation and compliance [5]. Vascular age is a means of expressing the biological age of a patient equivalent to their estimated 10-year CV risk. The concept of vascular age characterizes the gradual deterioration of vascular structure and function on an individual level, as opposed to chronological age that simply refers to the passage of time [6]. The measurement of arterial age and its relation to chronological age allows practitioners to assess the presence of early vascular ageing at an individual level [7].

Although vascular age is a comprehensible way to report individualized risk to patients, the concept of arterial age as a surrogate marker of CV risk is a relatively novel idea [8]. A number of standardized, non-invasive methods and approximations have recently become available for the calculation of vascular age, including those derived from the Framingham Risk Score (FRS) and the Systematic Coronary Risk Evaluation (SCORE) [9,10]. Both methods, however, have shown to display a tendency of overestimating CVD risk in contemporary populations [10,11,12]. Aside from traditional CV risk models, several promising techniques have emerged for the calculation of arterial age, including the ultrasonographic assessment of carotid intima-media thickness (CIMT) and measurement of pulse wave velocity (PWV) [13,14]. Furthermore, previous evidence has suggested the reliability of defining vascular age based on data derived from coronary artery calcium score (CACS) [15]. Direct visualization of atherosclerosis yields the potential to better define coronary heart disease (CHD) risk by detecting manifest atherosclerotic damage and when compared with traditional risk scores, this technique has demonstrated an increased sensitivity in detecting of CHD risk [16]. The concept of deriving vascular age from CACS measurements is to estimate the age at which the estimated CHD risk is equivalent to the observed CACS [17].

While FRS, SCORE, and CACS have all been utilized to measure individual vascular age previously, data comparing these techniques are scarce. In this study we aimed to compare vascular age calculations based on FRS, SCORE, and CACS. We hypothesized that vascular age calculations based on traditional risk factor analyses could potentially display significant differences to the CACS-derived methodology, which may impact CV preventive strategies.

## 2. Materials and Methods

### 2.1. Study Populatioan

In our cross-sectional single center study, 241 low and intermediate CHD risk Caucasian patients with stable chest pain referred to clinically indicated coronary CT angiography (CCTA) were consecutively enrolled between June 2019 and August 2021. Blood samples were drawn from patients as part of the Molecular Fingerprinting in Coronary Heart Disease trial. Briefly, the trial primarily focuses on identifying specific molecular patterns with infrared spectroscopy that are associated with CHD. Patients between 40 and 75 years who consented to participate in this study met the inclusion criteria. Patients with previous coronary intervention or coronary bypass operation were excluded from the trial. Due to their potential interfering effect on molecular analyses, further exclusion criteria included the presence of current/previous malignancies, serious renal/hepatocellular insufficiency, autoimmune disease, and atrial fibrillation. Prior to the CCTA examination, detailed demographic, anthropometric and medical history of all patients were recorded. Hypertension was defined as the regular administration of antihypertensive agents, while the definition of dyslipidemia relied on the regular, documented prescription of statins. Blood pressure and heart rate were measured twice, once an hour before the CT examination and once directly preceding the scan using a validated oscillometric device (Omron M3, Kyoto, Japan). The average of the two measurements is reported. A comprehensive blood panel including total cholesterol, low-density lipoprotein (LDL) cholesterol, and high-density lipoprotein (HDL) cholesterol was carried out at enrollment. Once the results of the blood test were available, calculations of vascular age derived from different CV scores were performed.

### 2.2. Coronary Computed Tomography Angiography

All patients underwent axial non-contrast enhanced coronary artery scans and coronary CT angiography (CCTA) using a 256-slice CT scanner (Philips Brilliance iCT, Best, The Netherlands) or a dedicated cardiovascular scanner (CardioGraphe, GE Healthcare, Chicago, IL, USA). Images were acquired during a single breath-hold, using prospective ECG-triggering. Images were reconstructed using iterative reconstruction algorithms with 0.4 mm slice thickness (iDose, Philips Healthcare, Cleveland, OH, USA/ASiR, GE Healthcare, Waukesha, WI, USA). The amount of coronary calcium on the non-contrast-enhanced images was quantified using the Agatston scoring method [18]. The evaluation of the extent, severity and distribution of coronary artery disease (CAD) was assessed by experienced readers (with 5–10 years of experience in cardiac CT). We defined severe CAD as the presence of significant luminal diameter stenosis (≥70% or ≥50% in case of the left main coronary) in ≥1 major coronary artery.

### 2.3. Calculation of Vascular Age Based on Coronary Artery Calcium Score

A previously described method was used to estimate the arterial age of the patients [17]. Arterial age of an individual can be expressed by equating estimated CHD risk for observed age and CAC. The arterial age of a participant, that is the age at which the estimated CHD risk is the same as that for the observed CACS, can be calculated with a mathematical equation: arterial age = 39.1 + 7.25 × log_10_(CACS + 1). The difference in vascular and chronological age was calculated subsequently. A positive difference indicates that a subject’s 10-year CHD risk is higher than what would be expected given the documented chronological age (vascular age + group), while a negative difference implies that a participant’s 10-year CHD risk is less than that applying his or her chronological age (vascular age — group). Patients whose chronological age was equal to their corresponding biological age were assigned to the vascular age–subgroup.

### 2.4. Calculation of Vascular Age Based on Framingham Risk Score (FRS)

For FRS-derived vascular age calculation, we applied a method that has previously been described by D’Agostino et al. [10]. The proposed risk model incorporates age, total and HDL cholesterol, systolic blood pressure (SBP), ongoing treatment of hypertension, smoking, and diabetes. This original paper introduced the concept of vascular age by transforming the CVD risk of an individual to the age of a person with the same risk but all other risk factors at the normal level.

### 2.5. Calculation of Vascular Age Based on Systematic Coronary Risk Evaluation (SCORE) Risk Score

The SCORE project equations encompass age, gender, smoking status, total cholesterol and SBP levels [19]. The concept behind SCORE-derived vascular age calculations rely on the same principles as the concept proposed by D’Agostino et al. Vascular age of a patient with CV risk factors is defined as the age that an individual of the same sex as the given patient should be if he/she had the same absolute risk but controlled risk factors [9].

### 2.6. Statistical Analysis

Descriptive data are expressed as mean ± standard deviation or median with interquartile ranges or percentages as appropriate. The normality of continuous parameters was tested with Kolmogorov-Smirnov test. Differences between descriptive characteristics, laboratory parameters, CACS, and different CV risk-based vascular age derivates were compared between groups using unpaired Student’s *t*-tests or Mann–Whitney rank-sum test for data failing tests of normality, while chi-square test was applied for categorical variables. The Wilcoxon signed-rank test was performed in order to compare vascular ages measured by the different risk scores in the whole cohort and in the studied subgroups (patients with hypertension and patients with diabetes). For the comparison of the distribution of the proportion of subjects with lower or higher vascular age based on CACS, FRS, and SCORE, the McNemar’s test was used, while the strength of agreement was assessed by Cohen’s kappa. Intraclass correlation coefficients were calculated, and Bland-Altman plots were generated to assess agreement between pairs of risk estimates. Pearson correlation was performed to assess correlation between different score-based vascular ages. A two-sided *p* < 0.05 was considered to be significant in all of the analyses. SPSS (Armonk, NY, USA version 25.0) was used for all calculations. 

## 3. Results

A total of two hundred and forty-one patients were enrolled in our study. The median age of our cohort was 56.2 (IQR: 48.5–66.2) years, approximately half of them were female (117/241, 48.5%) and had dyslipidemia (117/241, 48.5%), while the median calcium score was 2.0 (IQR: 0.0–99.0). Demographic parameters, clinical data, and laboratory results are summarized in Table 1 for the entire cohort and for the groups of subjects whose vascular age was higher than their chronological age with the three used methods, as well (CACS+, FRS+, SCORE+). The median of FRS and SCORE-based vascular ages were 68.0 (IQR: 55.0–82.0) and 63.0 (IQR: 53.0–75.0) years, respectively, which resulted in the addition of 11.8 and 6.8 years to the median chronological age (both *p* < 0.001). On the other hand, CACS-adjusted median vascular age (47.1 years) did not significantly differ from the median chronological age (*p* = 0.19). Overall, the CACS+ group demonstrated significantly higher calcium score, and CACS-derived vascular age than the FRS+ and SCORE+ subpopulations (both *p* < 0.001). The proportion of females was significantly higher in the SCORE+ group as compared to CACS+ patients (*p* = 0.01). Moreover, the SCORE+ group displayed a higher rate of hypertension (*p* = 0.03) and dyslipidemia (*p* = 0.02) than the CACS+ population.

Overall, 11.2% (27/241) of patients had severe CAD in CCTA. Median vascular ages proved to be significantly higher calculated by CACS (80.0 [IQR: 65.0–94.2] vs. 39.1 [IQR: 39.1–68.4], *p* < 0.001) and FRS (80.0 [IQR: 64.0–84.0] vs. 68.0 [IQR: 55.0–82.0], *p* = 0.03) in patients with significant stenosis, while no association was delineated for FRS. Results of vascular age calculations in patients with and without significant stenosis are detailed in Appendix A.

Figure 1 demonstrates the comparison of the different score-based vascular ages. Regarding the entire cohort, no statistical difference was observed between FRS- and SCORE-derived vascular ages (*p* = 0.08), while the CACS-adjusted biological age was significantly lower than the other 2 age derivates (both *p* < 0.001, Figure 1A). Figure 1B demonstrates the pairwise comparisons of the FRS, SCORE, and CACS vascular ages in hypertensive patients (*n* = 151). All of the comparison combinations were significant, as FRS-adjusted biological age was significantly higher than SCORE and CACS vascular ages and CACS-adjusted age was significantly lower than FRS and SCORE biological ages (all *p* < 0.001). Regarding patients with documented diabetes (*n* = 33), vascular age calculated by FRS remained statistically higher than SCORE- and CACS-derived vascular ages (both *p* < 0.001), while median biological age measured by SCORE and CACS did not differ significantly (*p* = 0.16, Figure 1C). 

Vascular ages obtained from SCORE and FRS showed excellent correlation [ICC: 0.91 (95%CI: 0.88–0.93)]. Although all pairwise combinations proved to be statistically significant (all *p* < 0.001), the correlation of CACS-derived vascular ages was only moderate when compared with biological ages measured by FRS [ICC: 0.66 (95%CI: 0.56–0.73)] and SCORE [ICC: 0.65 (95%CI: 0.56–0.73)]. The pairwise comparisons and the corresponding ICC values are depicted on Figure 2. The differences were then examined using Bland-Altman plots (Figure 3). The comparison of SCORE and FRS demonstrated the highest agreement, with an overestimation of vascular age by a mean of 4.2 years using the SCORE-based models (Figure 3A), while on average, the CACS-based model measured 12.2 and 8.1 years less than the FRS (Figure 3B) and SCORE (Figure 3C) equations. All of the pairwise combinations proved to be statistically significant (all *p* < 0.001).

Depending on the relation of their arterial age and the corresponding chronological age, patients were classified either as vasc age + (vascular age > chronological age) or vasc age − (vascular age ≤ chronological age) for all three methods (Figure 4). Overall, 92/241 (38.2%) of the patient cohort was found to be vasc age + with all 3 methods. On the other hand, only 7/241 (2.9%) patients were found to have a healthy vasculature and were therefore classified as vasc age–with all 3 methods. The overall agreement between the different methods proved to be fair in the case of SCORE-FRS comparison (κ = 0.30), while only slight agreements appeared for FRS-CACS and SCORE-CACS comparisons (κ = 0.13 and κ = 0.02, respectively). When compared with the FRS-adjusted vascular age groups, the utilization of CACS resulted in the reclassification of 115/241 (47.7%) patients in total. Overall, 107/201 (53.2%) subjects previously classified as vasc age + by FRS were assigned to the vasc age–group with CACS-based calculations (*p* < 0.001). When results of the SCORE- and CACS-derived classification of individual patients were compared, adjusted vascular age classification was discordant in 134/241 (55.6%) cases, including 129/226 (57.1%) subjects who were classified as vasc age + by SCORE but were reclassified as vasc age–by CACS-based calculations (*p* < 0.001).

## 4. Discussion

Based on its ability to detect manifest CAD, CACS yields the potential of extending the CV risk assessment of patients beyond traditional risk models. Available data regarding the comparison of CACS- and traditional risk score-derived vascular ages are, however, scarce in patients undergoing clinically indicated CCTA. In our current cross-sectional study, we compared vascular ages calculated with traditional risk scores (FRS and SCORE) and CACS in a cohort of low and intermediate CHD risk patients with stable angina. Substantial differences were demonstrated between vascular ages measured by CACS and conventional risk scores, however, discrepancies between FRS- and SCORE-derived biological ages became more prominent when assessing subgroups of patients with treated hypertension and diabetes. Overall, traditional methods of vascular age calculation displayed a tendency of overestimating vascular age as compared with CACS.

The mathematical concept of absolute CV risk is, for the majority of the patients, difficult to interpret. The concept of vascular age represents the idea that chronological age does not truly reflect the status of their vascular aging process and provides patients valuable information about the relationship between the age of their vasculature and their actual age. CV risk estimates provided in a more comprehensible way potentiate an improved adherence to drug treatment and lifestyle modifications recommended by the clinicians [20]. However, although the identification of patients with early vascular aging would be of outmost importance to initiate adequate prevention strategies, a substantial discordance is documented between different CV risk-based vascular age calculations [21], which was also confirmed by our results. This fact could lead to a mismatch in the selection of patients identified by different vascular age calculation methods leading to the decreased effectiveness of preventive strategies. Detecting manifest coronary atherosclerosis carries the potential of more accurately defining the true CV risk of individuals by direct vascular imaging [13,22]. Additionally, CTA-based pictorial information targeting patients could facilitate the perception of the link between modifiable risk factors and atherosclerosis and, thus, may be helpful to further increase the adherence of patients to lifestyle changes and pharmaceutical treatment, as compared to traditional risk score models. Although our findings have major implications for individual decision-making on preventive therapy, the direct comparison of the effectiveness of CV risk score-based and measurement-based vascular age calculation methods is warranted in prospective studies to define the most effective, “gold standard” method.

Although the correlation of FRS- and SCORE-based calculations proved to be excellent (and statistically significant) with a relatively high level of agreement, it does not imply that they lead to identical results. When assessing the presence of accelerated vascular aging (vascular age > chronological age), approximately 15% (35/241) of the patients were reclassified between the models. It is safe to assume that the observed differences between conventional risk score-derived vascular ages are driven by methodological differences. CV risk estimation by SCORE is known to be driven primarily by age, meaning that a young patient with a series of CV risk factors may still be assigned to a low absolute risk [9]. Moreover, original SCORE CV risk calculating equations consider 10-year risk for CV mortality, while FRS, on the other hand, integrates comorbidities such as diabetes and treated hypertension into the model and predicts CV events plus mortality [10,19]. It is plausible that the observed, more pronounced differences between traditional risk score-derived vascular ages in chronic hypertensive and diabetic patients are explained by this fact and FRS-based vascular age calculations may allow for a more precise risk stratification of patients in these cohorts.

Expressing arterial age as a surrogate marker for atherosclerotic burden in terms of CHD risk prediction may allow for a more precise risk assessment of patients with stable angina. Theoretically, a potential explanation for conventional risk score-based techniques underestimating vascular age can be that they integrate factors (e.g., heart failure) that could occur even without concomitant CAD [23]. Prospective studies are warranted to determine whether this lower risk in CACS-based derivates potentiate the implementation of more adequate preventive strategies in this patient cohort.

The present study has limitations that should be considered. We solely enrolled patients that were referred to CCTA due to stable angina, which limits the generalizability of our results. Furthermore, since the described vascular age equations were developed more than a decade ago in cohorts which differed from the cohort of the present study, it is plausible that these methodologies cannot necessarily be implemented in contemporary populations, however, no restriction has been defined to avoid the used vascular age calculation methods in different patient populations. Given that race-specific differences are well-documented in the pattern of CV risk, the fact that all of the enrolled patients were Caucasian is a shortcoming. Although the FRS and SCORE cohort is relatively homogenous regarding patient demographics (predominantly Caucasian patients), the Multi-Ethnic Study of Atherosclerosis, from which CACS-based calculations were derived, incorporates a well-balanced ethnic diversity allowing wider generalizability. By enrolling patients with more diverse demographics in future studies similar to ours, valuable insights could be provided to aid race-specific implementation of the conclusions. Additionally, evaluation of blood pressure was not executed according to current guidelines, as it was measured only on one arm, which could influence SCORE and FRS vascular age calculations. However, blood pressure was measured in a silent room by an assistant after more than 5 min resting position which limited white-coat effect. When compared with conventional risk models, the superiority of CACS-based vascular age assessment was not demonstrated in the current study as the incremental effect of this model on overall CV morbidity and mortality needs to be clarified with prospective studies in the future.

## 5. Conclusions

Our results demonstrated that vascular age on the basis of CACS displays marked differences from conventional risk score-based methods in patients referred to clinically indicated CCTA. With further prospective studies supporting our findings, CACS-adjusted vascular age calculation models may emerge as a tool that allows for more effective preventive interventions in low- to intermediate-risk patients with stable angina.

## Figures and Tables

**Figure 1 jcm-11-01111-f001:**
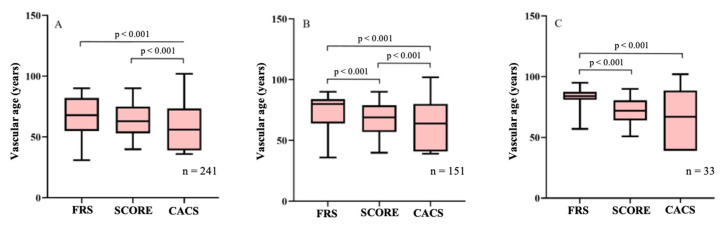
Box-and-whiskers plots demonstrating the difference between vascular age calculated with FRS, SCORE, and CACS. Wilcoxon signed-rank test was used to compare vascular ages measured by the different risk scores in the whole cohort (**A**) and separately focusing on patients with hypertension (**B**) and diabetes (**C**). (**A**). Significant differences could be observed between FRS and CACS (*p* < 0.001) and between SCORE and CACS (*p* < 0.001) in the full cohort. No difference was noted between FRS- and SCORE-derived vascular age (*p* = 0.08). (**B**). All comparison combinations proved to be significant when assessing patients with treated hypertension (all *p* < 0.001). (**C**). Regarding the cohort of patients with diabetes, vascular age calculated by FRS was significantly higher than other vascular age derivates (both *p* < 0.001), while SCORE- and CACS-derived vascular ages did not differ significantly (*p* = 0.16). FRS, Framingham Risk Score; SCORE, Systematic Coronary Risk Evaluation; CACS, coronary artery calcium score.

**Figure 2 jcm-11-01111-f002:**
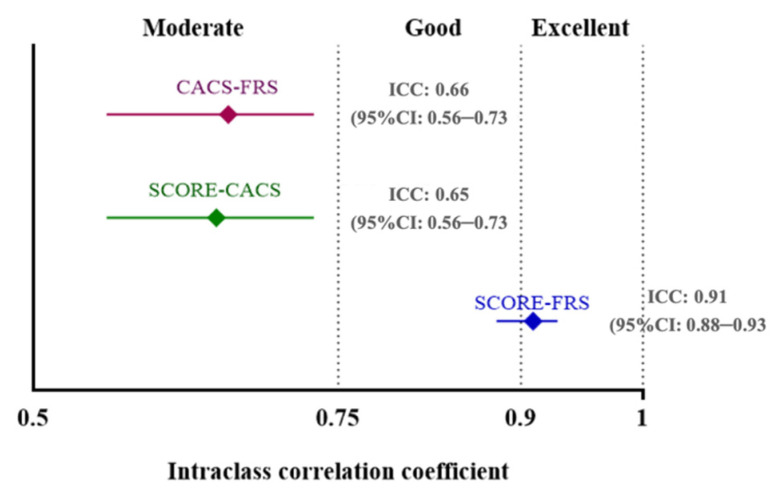
Intraclass correlation coefficients for the comparison of different risk score-based models. Intraclass correlation coefficient was calculated to assess correlation between different score-based vascular ages. While SCORE and FRS displayed excellent correlation, CACS only moderately correlated with FRS and SCORE. All pairwise combinations proved to be statistically significant (all *p* < 0.001). CACS, coronary artery calcium score; FRS, Framingham Risk Score; SCORE, Systematic Coronary Risk Evaluation.

**Figure 3 jcm-11-01111-f003:**
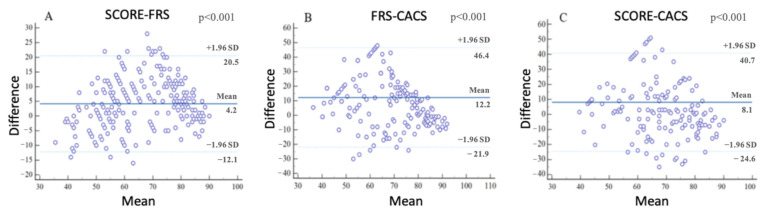
Bland-Altman plots demonstrating the differences between the risk equations. Compared with FRS, SCORE overestimated vascular age by a mean of 4.2 years (**A**), while on average, the CACS-based model measured 12.2 and 8.1 years less than the FRS (**B**) and SCORE (**C**) equations. All pairwise combinations proved to be statistically significant (all *p* < 0.001). SCORE, Systematic Coronary Risk Evaluation; FRS, Framingham Risk Score; CACS, coronary artery calcium score.

**Figure 4 jcm-11-01111-f004:**
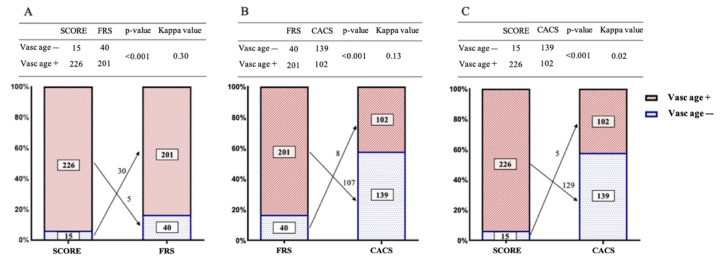
The distribution of patients with elevated and healthy vascular age using the different models. Patients demonstrating higher score-derived biological age than chronological age were classified as vasc age +, while patients with lower chronological age than their respective chronological age were labeled as vasc age –. Four-fold tables are provided to delineate the distribution of patients in these categories using the different models, as well as Cohen’s Kappa values describing the strength of agreement between them. While all pairwise comparisons appeared to be significant (*p* < 0.001), only the SCORE- and FRS-derived vascular age categories demonstrated fair agreement (κ = 0.30) (**A**), while FRS-CACS (**B**) and SCORE-CACS (**C**) comparisons resulted in only slight agreement (κ = 0.13 and κ = 0.02, respectively). The column charts provide visual depiction of the distributions, while the numbers on the arrows demonstrate the number of the reclassified subjects. McNemar’s test was used to display statistical differences in the distribution of patients in all pairwise comparisons (all *p* < 0.001). SCORE, Systematic Coronary Risk Evaluation; FRS, Framingham Risk Score; CACS, coronary artery calcium score.

**Table 1 jcm-11-01111-t001:** Demographic parameters and clinical data of the study participants.

	All Subjects	CACS+	FRS+	SCORE+
Number, (%)	241	102 (42.3)	201 (83.4)	226 (93.8)
Demographics				
Chronological age (years)	56.2 [48.5–66.2]	60.9 [50.2–68.1]	57.3 [48.7–66.8]	57.7 [48.5–66.4]
CACS vascular age (years)	47.1 [39.1–72.3]	75.7 [66.3–84.8]	55.8 [39.1–75.9] *	50.0 [39.1–72.5] *
FRS vascular age (years)	68.0 [55.0–82.0]	80.0 [60.0–84.0]	76.0 [60.0–83.0]	72.0 [57.0–82.0]
SCORE vascular age (years)	63.0 [53.0–75.0]	69.5 [57.0–78.3]	65.0 [54.5–77.0]	64.0 [54.0–76.0]
Female sex, *n* (%)	117 (48.5)	34 (33.3)	87 (43.3)	109 (48.2) *
BMI (kg/m^2^)	27.2 [24.7–30.5]	28.4 [25.4–31.7]	27.6 [24.9–31.1]	27.2 [24.7–30.5]
Cardiovascular risk factors				
Current smoker, *n* (%)	38 (15.8)	21 (20.6)	36 (17.9)	38 (16.8)
Hypertension, *n* (%)	151 (62.7)	75 (73.5)	141 (70.1)	139 (61.5) *
Diabetes mellitus, *n* (%)	33 (13.7)	18 (17.6)	33 (16.4)	32 (14.2)
Dyslipidemia, *n* (%)	117 (48.5)	63 (61.8)	102 (50.7)	109 (48.2) *
SBP (mmHg)	146.5 ± 18.4	149.1 ± 17.8	149.8 ± 17.6	148.3 ± 17.6
DBP (mmHg)	88.9 ± 10.4	89.9 ± 10.5	90.1 ± 10.2	89.6 ± 10.1
Laboratory parameters				
Glucose (mmol/L)	5.4 [5.1–5.8]	5.5 [5.2–6.1]	5.4 [5.1–5.9]	5.4 [5.1–5.8]
GFR (mL/min/1.73^2^)	78.5 [68.2–87.0]	75.2 [67.6–87.0]	78.5 [68.5–87.0]	78.2 [68.2–87.6]
Total cholesterol (mmol/L)	5.0 [4.2–5.9]	5.0 [4.0–6.1]	5.1 [4.2–6.1]	5.1 [4.3–6.0]
LDL-cholesterol (mmol/L)	3.3 [2.4–4.0]	3.1 [2.3–4.2]	3.4 [2.4–4.2]	3.3 [2.4–4.1]
HDL-cholesterol (mmol/L)	1.3 [1.1–1.7]	1.3 [1.1–1.7]	1.3 [1.1–1.6]	1.3 [1.1–1.7]
Triglyceride (mmol/L)	1.4 [1.0–2.3]	1.5 [1.0–2.3]	1.5 [1.1–2.4]	1.5 [1.0–2.3]
Agatston score	2.0 [0.0–99.0]	154.5 [41.5–544.8]	9.0 [0.0–154.5] *	3.5 [0.0–103.3] *

Different risk-derived vascular ages were compared using Mann-Whitney U-test. CACS+, subjects with elevated vascular age based on coronary artery calcium score; FRS+, subjects with elevated vascular age based on Framingham Risk Score method; SCORE+, subjects with elevated vascular age based on Systematic Coronary Risk Evaluation method. CACS, Coronary artery calcium score; FRS, Framingham Risk Score; SCORE: Systematic COronary Risk Evaluation method; BMI, Body mass index; SBP, Systolic blood pressure; DBP, Diastolic blood pressure; GFR, Glomerular filtration rate; LDL, Low density lipoprotein; HDL, High density lipoprotein. Continuous variables are expressed as mean ± standard deviation (SD) or median and interquartile range (IQR), while categorical variables are expressed as numbers and percentages. Significant differences are signed as bold and italic characters: * significant difference from CACS+.

## Data Availability

The data presented in this study are available on request from the corresponding author. The data are not publicly available due to reasons pertaining to patient privacy.

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
