# Peer review of "Correlation between Coronary Artery Calcium- and Different Cardiovascular Risk Score-Based Methods for the Estimation of Vascular Age in Caucasian Patients"

_jcm, 2022, doi:10.3390/jcm11041111_

Round 1
Reviewer 1 Report
- The authors refer to references of McCelland et al. D'Agostino et al. and Conroy et al. whre the caluclations of the CACS, FRS and SCORE are described. However, these papers have been published some years ago (in 2009, 2008 and 2003). How can they be sure that the described methods are still valid in their study population?
- In order to compare the proportions of subjects with lower or higher vasculkar age between two methods (e.g. FRS and CACS) McNemar test seems to be appropriate (instead of Cochran's Q).
- Results, table 1: The difference between the medians of chronological age and FRS vascular age is 11.8; compared to the SCORE vascular age the diffence is 6.8 years. However, in the text differences of 16.8 and 11.8 are presented.
- In Figures 1 the sample sizes should be given.
- Figures 3 and the text before are rather difficult to understand. In my opinion, a table with the combinations of the Scores results and the corresponding frequencies would be helpful. Another possibility would be the presentation of forufold tables in order to compare to score methods. Kappa coefficients woulrd be useful as a measure of agreement.
- Figure 2: The correlations in Figure 2a and 2b are statistically significant. However, this does not mean that the FRS and SCORE lead to identical results. This issue should be addressed. The intraclass correlation coefficient is more adequate in order to assess method agreement.
Author Response
1. The authors refer to references of McCelland et al. D'Agostino et al. and Conroy et al. where the calculations of the CACS, FRS and SCORE are described. However, these papers have been published some years ago (in 2009, 2008 and 2003). How can they be sure that the described methods are still valid in their study population?
The Reviewer has addressed a major limitation of the current study that should be highlighted in the manuscript. Although the SCORE project dates back to 2003, the method of SCORE-based vascular age calculation was published in 2010 by Cuende et al. Nevertheless, it is plausible that the methodologies described are not necessarily implementable in current populations more than a decade later. Our assumption should be confirmed by further prospective validation studies in contemporary populations.
The limitations section was supplemented with the following sentence: “Furthermore, since the described vascular age equations were developed more than a decade ago in cohorts which differed from the cohort of the present study, it is plausible that these methodologies cannot necessarily be implemented in contemporary populations, however, no restriction has been defined to avoid the used vascular age calculation methods in different patient populations."
2. In order to compare the proportions of subjects with lower or higher vascular age between two methods (e.g. FRS and CACS) McNemar test seems to be appropriate (instead of Cochran's Q).
We thank Reviewer #1 for the insightful statistical comment. We recalculated each part accordingly and all comparison combinations remained statistically significant (p<0.001) using McNemar’s test.
3. Results, table 1: The difference between the medians of chronological age and FRS vascular age is 11.8; compared to the SCORE vascular age the difference is 6.8 years. However, in the text differences of 16.8 and 11.8 are presented.
We are grateful for Reviewer #1 for pointing out the error in the text. We can confirm that the data in Table 1 is correct, the typographical errors were corrected in the manuscript.
4. In Figures 1 the sample sizes should be given.
We completely agree that it is insufficient for the sample sizes to only appear in Table 1. Currently, Figure 1 contains the relevant sample sizes, while the text was also supplemented with the corresponding numbers.
5. Figures 3 and the text before are rather difficult to understand. In my opinion, a table with the combinations of the Scores results and the corresponding frequencies would be helpful. Another possibility would be the presentation of fourfold tables in order to compare to score methods. Kappa coefficients would be useful as a measure of agreement.
We would like to thank Reviewer #1 for the insightful suggestion. The figure (now Figure 4) was supplemented with the results of the FRS-SCORE comparison, as well as four-fold tables detailing the corresponding frequencies and the calculated Kappa-values. We are convinced that these additions raise the clarity and the overall quality of the manuscript. The Figure 4 legend, as well as the Statistical analysis and Results paragraph was modified accordingly.
6. Figure 2: The correlations in Figure 2a and 2b are statistically significant. However, this does not mean that the FRS and SCORE lead to identical results. This issue should be addressed. The intraclass correlation coefficient is more adequate in order to assess method agreement
We are grateful for the suggestion, intraclass correlation coefficient has been calculated between the different scores instead of Pearson correlation. The results of the analysis in all pairwise combinations are presented on Figure 2.
The abstract and the Statistical Analysis paragraph was modified accordingly, while the Results was supplemented with the following sentence: “Although the correlation of FRS- and SCORE-based calculations proved to be excellent (and statistically significant) with a relatively high level of agreement, it does not imply that they lead to identical results. When assessing the presence of accelerated vascular aging (vascular age > chronological age), approximately 15% (35/241) of the patients were reclassified between the models.”
The figure containing the results of the Pearson analysis is currently provided as Supplementary material.
Reviewer 2 Report
The authors estimate vascular age by use of various formulae in a group of 241 patients who had chest pain and clinically indicated CT angiography workup. They concluded that vascular age calculated based on Framingham or SCORE risk parameters and over- or underestimate vascular age calculated based on coronary artery calcium (CAC).
A major concern surrounds the validity of comparing vascular age without any direct measurements or a gold standard. Vascular age was not measured (ie as traditionally done by CIMT or pulse wave doppler) in this cohort. Instead, vascular age was calculated using CAC and a formula derived from the MESA cohort. It’s not clear that the MESA formula would apply to the population described in this study. These are 2 very different populations with MESA consistently of subjects who were asymptomatic, low risk, and ethnically diverse. In contrast, the patients in this study all had chest pain requiring further workup and were all Caucasian. Ultimately, what’s being compared in this manuscript is calculated CVD risk assessed by CAC vs risk scores. While no outcomes were described in this study, it is already well known that CAC improves risk prediction in many populations.
Author Response
A major concern surrounds the validity of comparing vascular age without any direct measurements or a gold standard. Vascular age was not measured (ie as traditionally done by CIMT or pulse wave doppler) in this cohort. Instead, vascular age was calculated using CAC and a formula derived from the MESA cohort. It’s not clear that the MESA formula would apply to the population described in this study. These are 2 very different populations with MESA consistently of subjects who were asymptomatic, low risk, and ethnically diverse. In contrast, the patients in this study all had chest pain requiring further workup and were all Caucasian. Ultimately, what’s being compared in this manuscript is calculated CVD risk assessed by CAC vs risk scores. While no outcomes were described in this study, it is already well known that CAC improves risk prediction in many populations.
We agree with Reviewer #2, that in our study the calculated Framingham and SCORE vascular ages were not compared with any "gold standard" method of vascular age measurement. However, there is no international consensus available defining the gold standard method of vascular age calculation. We suppose, that the evaluation of coronary calcium content and the visualization of carotid artery intima-media thickness (CIMT) are similar considering vascular properties, furthermore, CAC score has more evidence for CV outcome prediction, so we disagree that CIMT-based vascular age calculation would be superior than CAC score-based method. It would require further studies to clarify this issue, but in the study of Khalil J et al (doi: 10.1111/j.1751-7141.2010.00071.x) the evaluated vascular ages with the two methods correlated well.
We also agree, that our population is different from MESA, Framingham and SCORE populations, and we mention it now in the limitations as well, however, no restriction has been defined to avoid the used vascular age calculation methods in different patient populations.
Our study reveals the problem that different methods for the calculation of the similarly called "vascular age" can result in different outcome numbers which could cause bias in the preventive strategies. Our results warrant the definition of a "gold standard" method.
Reviewer 3 Report
The paper probably would be interesting for the readers, especially for readers who desire to focus on cardiovascular risk score models. I congratulate the authors, the research topic is original (comparison between cardiovascular risk scoring models / vascular age), comprehensible and fluent.
I personally appreciated the statistical methodology despite the conclusions of the study are partial (according to the structure and typology) but justified by the data.
I would ask the authors to send the raw tables with the analysis summary table and an English version of the “not published material”.
I would focus on the following aspects (n°1 & 2 considered as major):
- Figure 3: the description of figure 3 is not fully clear. Add graphic A (FRS vs SCORE) despite the clearly expected results in order to make the figure complete. On the other hand you can delete the letter A (always FRS vs SCORE) from the description of the figure.
- In order to improve the attractiveness of the paper I would recommend to further discuss the hypothetical clinical implications derived by CV risk under/overestimation (with several models) in term of outcomes/preventive therapy/approach.
- Please add the statistical tests used in every figure description, not only in the test.
- It would be interesting, using appropriate models or universal comparison tools, to report the expected differences of risk predicted between the 3 models (FRS, SCORE, tomography). I would insert this description in the result before the figure 3 (so before figure 3 where the focus is on reclassification percentage) or directly in the discussion. I appreciate that authors used specific terminology focusing on vascular age, at the same time a focus on risk estimation could improve the interest of readers: the discrepancy between models (expressed in term of risk and possibly in term of event expected) could be a good starter to discuss the clinical related implications (in the discussion session, clearly).
- As mentioned by authors, given the specific topic (risk models), the fact that all patients enrolled are caucasian is a weakness that should be better underlined. It could be useful to add that in the title.
- If available further data (tomography with contrast in the setting of CCS -chest pain-, n° patients undergoing angiography, n° ps undergoing revascularization ecc) please consider to discuss and analyze the risk models value in a new paragraph.
Author Response
1. Figure 3: the description of figure 3 is not fully clear. Add graphic A (FRS vs SCORE) despite the clearly expected results in order to make the figure complete. On the other hand, you can delete the letter A (always FRS vs SCORE) from the description of the figure.
We would like to thank Reviewer #3 for the great comment, changes to Figure 3 (now Figure 4) were implemented accordingly. The depiction of the FRS-SCORE comparison was added, along with four-fold tables detailing the distribution of patients, with the corresponding Kappa-values. We are entirely convinced that these changes improve our manuscript substantially. The figure legend, as well as the Statistical analysis and Results paragraph was modified accordingly.
2. In order to improve the attractiveness of the paper I would recommend to further discuss the hypothetical clinical implications derived by CV risk under/overestimation (with several models) in term of outcomes/preventive therapy/approach.
According to the insightful suggestion of Reviewer #3, we have supplemented the Discussion with the following paragraph:
“However, although the identification of patients with early vascular aging would be of outmost importance to initiate adequate prevention strategies, a substantial discordance is documented between different CV risk-based vascular age calculations [21], which was also confirmed by our results. This fact could lead to a mismatch in the selection of patients identified by different vascular age calculation methods leading to the decreased effectiveness of preventive strategies. Detecting manifest coronary atherosclerosis carries the potential of more accurately defining the true CV risk of individuals by direct vascular imaging [13, 22]. Additionally, CTA-based pictorial information targeting patients could facilitate the perception of the link between modifiable risk factors and atherosclerosis and, thus, may be helpful increase the adherence of patients to lifestyle changes and pharmaceutical treatment. Although our findings have major implications for individual decision making on preventive therapy, the direct comparison of the effectiveness of CV risk score-based and measurement-based vascular age calculation methods is warranted in prospective studies to define the most effective, "gold standard" method. ”
3. Please add the statistical tests used in every figure description, not only in the test.
All figure legends were augmented with information pertaining to the statistical test.
4. It would be interesting, using appropriate models or universal comparison tools, to report the expected differences of risk predicted between the 3 models (FRS, SCORE, tomography). I would insert this description in the result before the figure 3 (so before figure 3 where the focus is on reclassification percentage) or directly in the discussion. I appreciate that authors used specific terminology focusing on vascular age, at the same time a focus on risk estimation could improve the interest of readers: the discrepancy between models (expressed in term of risk and possibly in term of event expected) could be a good starter to discuss the clinical related implications (in the discussion session, clearly).
We agree that reporting the expected differences between the models provides valuable insight for the interpretation of the results and adds incremental value to our study. We have created three Bland-Altman plots as Figure 3 to depict the results of the pairwise comparisons and we discuss the corresponding results. The Statistical analysis, Results and Discussion paragraphs were supplemented accordingly.
While we acknowledge the potential importance of comparing the risk estimates of the different models, it would be incorrect methodologically, as they integrate different prognostic factors. The SCORE-based equation considers 10-year risk for cardiovascular mortality, while FRS and CACS-based models, on the other hand, further integrate cardiovascular events into the model aside from mortality. Accordingly, SCORE-based risk derivates would demonstrate a tendency to be significantly lower in comparison to the other models, while vascular age calculations provide a standardized method of assessment.
5. As mentioned by authors, given the specific topic (risk models), the fact that all patients enrolled are Caucasian is a weakness that should be better underlined. It could be useful to add that in the title.
This is one of the major limitations of the current study, which we also believe should be emphasized more markedly. Because of the relatively homogenous demographics of the Framingham and SCORE cohort consisting of predominantly Caucasian patients, we presumed that these models can be applied in the current population. However, by adapting these models we inherit their great weakness as well: the results of the FRS and SCORE-based calculations are most applicable to whites and may be inaccurate when applied to other ethnic groups, however, no strict limitation of the usefulness of the different methods on different races has been defined yet.
The title was modified accordingly, while the Limitations paragraph was amended: “Although the FRS and SCORE cohort is relatively homogenous regarding patient demographics (predominantly Caucasian patients), the Multi-Ethnic Study of Atherosclerosis, from which CACS-based calculations were derived, incorporates a well-balanced ethnic diversity allowing wider generalizability. By enrolling patients with more diverse demographics in future studies similar to ours, valuable insights could be provided to aid race-specific implementation of the conclusions.”
6. If available further data (tomography with contrast in the setting of CCS -chest pain-, n° patients undergoing angiography, n° ps undergoing revascularization ecc) please consider to discuss and analyze the risk models value in a new paragraph.
In spite of their excellent ability to predict 10-year cardiovascular mortality, the value of FRS and SCORE-based models to predict obstructive stenosis has previously been proven to be only moderate (doi: 10.1002/clc.22573), while CACS is a well-documented predictor of the presence of obstructive stenosis (doi: 10.1016/s0735-1097(00)01119-0). We agree that our current study provides further evidence to these documented correlations, however, we are not entirely convinced that these results produce a sufficient degree of novelty, especially considering that ICA was only done in a minority of patients.
Coronary CT angiography was performed for all patients besides the non-contrast enhanced scans and the results are available in our structured reporting platform. According to the suggestion of Reviewer #3, we have provided a supplementary table detailing the association of different risk model-based vascular ages and the presence of significant (>70%) stenosis, while also adding the following paragraph to the Methods: “Evaluation of the extent, severity and distribution of coronary artery disease (CAD) was assess by experienced readers (with 5-10 years of experience in cardiac CT). We defined severe CAD as the presence of significant luminal diameter stenosis (≥70% or ≥50% in case of the left main) in ≥1 major coronary artery.” Moreover, the Results was augmented with the following paragraph: “Overall, 11.2% (27/241) of patients had severe CAD in CCTA. Median vascular ages proved to be significantly higher calculated by CACS (80.0 [IQR: 65.0–94.2] vs. 39.1 [IQR: 39.1–68.4], p<0.001) and FRS (80.0 [IQR: 64.0–84.0] vs. 68.0 [IQR: 55.0–82.0], p=0.03) in patients with significant stenosis, while no association was delineated for FRS. Results of vascular age calculations in patients with and without significant stenosis are detailed in Table S1.
Round 2
Reviewer 1 Report
The paper has been thoroughly revised. All my comments and suggestions for improvement regarding efficeint data analysis and an adequate presentation of the results have been considered and implemented. I think that this has increased the quality of the paperconsiderably.
Author Response
We really appreciate the positive reception of our work and we are extremely grateful for all the valuable suggestions that increased the quality of our manuscript substantially!
Reviewer 2 Report
The limitations of this study are now better described.
Author Response
We are grateful for the compliment of Reviewer #2, his/her previous comments facilitated the improvement of our manuscript.
Reviewer 3 Report
The manuscript has been revised. The quality of paper is increased. Not all the changes/requests have been applied/answered.
Author Response
We thank Reviewer #3 for the positive feedback! We have thoroughly revised our manuscript and attempted to provide satisfactory answers and modifications according to his/her suggestions. Raw data, summary of analysis and all ‘not published material’ have now been uploaded along with the revised version of the manuscript.